# Effect of motivational interviewing intervention on HgbA1C and depression in people with type 2 diabetes mellitus (systematic review and meta-analysis)

**Kalayou Kidanu Berhe** [ORCID] *, **Haftu Berhe Gebru, Hailemariam Berhe Kahsay**

Mekelle University College of Health Sciences, School of Nursing, Mekelle, Ethiopia

\* Kalushaibex@gmail.com

**Data Availability Statement:** All relevant data are within the manuscript and its Supporting Information files. AND Study's data set (extracted from reviewed studies on effect of MI intervention

## Abstract

### Background

Many people living with diabetes are at risk for poor glycemic control, hyperlipidemia, hypertension, and macro vascular complications. Glycemic control and psychological wellbeing of the patient is mandatory for diabetes management. Addressing these issues in the early stages of the disease are the best interventions for decreasing poor glycemic control and psychological problems.

### Objective

To explore and analyze the literature for evidence of the effect of Motivational Interviewing (MI) intervention has on glycosylated hemoglobin A1C (HgbA1C) and depression in people with Type 2 diabetes mellitus (T2DM).

### Methods

A systemic review and meta-analysis of studies published in Cochrane Library, Google scholar, PubMed, & clinical trials.gov between 01/01/2009 and 12/06/2020 was performed. Inclusion criteria included RCT and pre post studies that assessed the effects of Motivational Interviewing on Hgb.A1C and depression in adults with T2DM. Weighted mean differences with 95% confidence intervals were calculated for continuous data. The data were synthesized and analyzed in a narrative form in systematic review and meta-analysis which was conducted using RevMan 5.2.0 & STATA version 11 software. Data were evaluated by weighted mean differences (WMDs) and 95% CIs.

### Result

Of the total identified 121 studies, eight were eligible for inclusion in the review. The pooled results showed that MI resulted in a significant improvement of mean HgbA1C level in the intervention group when compared with the control group (*WMD, -0.29; 95% CI, -0.47 to -0.10; p = 0.003, $I^2$ = 48%*). Effect of MI intervention on depressive symptoms was identified

on HgbA1C and depressive symptoms among T2DM patients) was uploaded as Supporting Information file 2 (Microsoft Excel speed sheet)

**Funding:** Authors received no specific funding for this work.

**Competing interests:** The authors have declared that no competing interests exist.

**Abbreviations:** BDI-II, Beck Depression Inventory-II and; CES-D, Center for epidemiologic studies depression scale; FBS, Fasting Blood Sugar; GADS, Generalized Anxiety Disorder Scale; HADS-D, Hospital Anxiety and Depression Scale-Depression; Hgb. A1C, Glycosylated hemoglobin A1C; MI, Motivation interviewing; NIDD, None Insulin Dependent Diabetes; NIDDM, None Insulin Dependent Diabetes Mellitus; PHQ-9, Patient Health Questionnaire-9; PICOS, Population, Intervention, Control, out come and Study design; PRISMA, Preferred Reporting Items for systematic Reviews and meta-analysis; RBS, Random Blood Sugar; RCTs, Randomized Controlled Trials; T2DM, Type 2 diabetes mellitus; WMDs, weighted Mean Differences.

through subgroup analysis according to intervention session time (30 or 60–80 minutes) and Follow-up period (3 or 24 months) then result showed that there was no significant difference in the reduction of depressive symptoms between the intervention and control groups. The output results were (WMD, -1.58; 95% CI, -5.05 to -0.188; p = 0.37; I2 = 48%), (WMD, -4.30; 95% CI, -9.32 to -0.73; p = 0.09; I2 = 95%), (WMD, -4.45; 95% CI, -10.58 to 1.69; p = 0.16; I2 = 96%) and (WMD, -2.12; 95% CI, -5.54 to 1.30; p = 0.22; I2 = 83%) respectively.

## Conclusion

The pooled result in meta-analysis indicated that motivational interviewing is effective in reducing HgbA1C but not depressive symptoms of patients with type 2 diabetes. Motivational interviewing intervention is important for diabetes management and effective in glycemic control with no effect on the reduction of depressive symptoms among persons with type 2 diabetes mellitus.

## Systematic review registration number

CRD42019146368.

## 1. Introduction

Diabetes is a group of metabolic diseases characterized by hyperglycemia resulting from defects in insulin secretion, insulin action, or both [1]. There are two major types of diabetes: Type 1 diabetes developed due to autoimmune β-cell destruction and type 2 diabetes which occurred due to a progressive loss of β-cell often on the background of insulin resistance. Type 2 diabetes accounts for 90–95% of those with diabetes [2]. The global diabetes prevalence in 2019 is estimated to be 9.3% (463 million people), rising to 10.2% (578 million) by 2030 and 10.9% (700 million) by 2045 [3].

Chronic hyperglycemia is associated with long-term damage, dysfunction and functional failure of the eyes, kidneys, nerves, heart, and blood vessels [4]. The increased occurrence and consequences of diabetes complications is alarming. The need for aggressive interventions which includes encourage patients to make positive changes in their health behaviors, teaching them how to better manage their diabetes [5].

Diagnosis of diabetes has significant impact on an individual's life, regular medication, frequent appointments and lifestyle changes that can lead to a number of emotional responses including diabetes related distress, depression, low mood, burnout, fear of injection or hypoglycemia, anxiety, eating disorders and problems with personal and sexual relationships [6]. Diabetes has an effect on mental health, health related quality of life, the impact of which not only affects the body but the persons' finance and health care system [7]. The bidirectional relationship between psychological problems and diabetes is also affected by earlier psychological problems and susceptibilities [8].

Diabetic patients with psychosocial problems often show negative coping strategies that negatively affect their future, resulting in increased diabetes fatalism (perceptions of despair, hopelessness and powerlessness), decreased medication adherence, and levels of self-care behaviors [9]. Psychosocial problems seem to be common among diabetic patients worldwide [10]. Study finding revealed that a person with diabetes was 2–3 times more likely to be

depressed than the person without diabetes [11]. Another study also revealed that the prevalence of depression was twice as high among people with diabetes (17.6%) when compared to those without diabetes (9.8%) [12].

The prevalence of anxiety and depression among patients with diabetes mellitus in India was found to be 56.43% [13]. Systematic review revealed that the prevalence of depression among diabetes patients in Ethiopia was 39.73% [14]. A study in Turkey showed that major depressive disorder was more frequent in diabetes patients with poor glycemic control than in those with good glycemic control [15].

Motivational interviewing (MI) is client based way of life style modification which is supported by scientific evidences for management of substance abuse and other long-term illness like diabetes. Its goal is to identify and decrease patient inconsistency in health related activities and promote patients' insight for advantage of healthy life style modification and self-reliance [16]. A recent review of Motivational Interviewing (MI) showed improvement in health behavior (e.g. diet and exercise) in patients with diabetes [17]. To the best of our knowledge, there has not been any systematic review and meta-analysis on effect of Motivational interviewing (MI) interventions on glycosylated hemoglobin A1C (HgbA1C) level and depression among patients with type 2 diabetes mellitus.

Past reviews focused on effect of Motivational Interviewing (MI) on some self-care practices domains among people with type 2 diabetes [5]. The impact of Motivational interviewing (MI) for T2DM patients [18] delivered by general physicians, Comparison of Motivational Interviewing (MI) effect delivered in different medical care settings [19], identify effective interventions for decreasing diabetes distress [20]. Motivational interviewing (MI) in the management of glycemic condition [21]. The effectiveness of Motivational Interviewing (MI) on blood glucose control among people with T2DM [22], impact of Motivational Interviewing (MI) on self-care practices among clients with insulin dependent diabetes mellitus [23], behavioral and drug related interventions for depression in patients with diabetes mellitus [24] and identifying psychosocial interventions that improve both physical and mental health in patients with diabetes [25].

It is broadly argued that resolving the psychosocial problems of people with T2DM could enhance psychological well-being, improve their quality of life and self-care practices, control the disease and reduce diabetes-related complications [9]. Moreover, evidence on these would encourage involvement of diabetes care professionals in Motivational Interviewing (MI) intervention for managing glycemic condition and depression in people diagnosed with DM. As the number of people with poor glycemic control and psychological disorder increase, it is imperative the suitable intervention to address those shortcomings be identified. Therefore, the purpose of this review is to examine and analyze empirical evidence for the effects of Motivational Interviewing (MI) intervention on glycosylated hemoglobin A1C (HgbA1C) level and depression in people with T2 DM.

## 2. Methods

### 2.1 Study question

First does Motivational Interviewing (MI) intervention decrease glycosylated hemoglobin A1C (HgbA1C) level? Second does Motivational Interviewing (MI) intervention have an effect in reducing depressive symptoms in people with type 2 diabetes mellitus? To answer the study questions, a systematic review and meta-analysis were undertaken using modified Cochrane method of systematic review of quantitative data [26]. The selection criteria for eligible studies and data extraction process were done based on the PICOS format and PRISMA guideline.

## 2.2 Types of studies

Studies with randomized controlled clinical trials (RCTs) or pre-post intervention were included.

## 2.3 Eligibility criteria

All globally conducted studies which are published between 01/01/2009 and 12/06/2020, peer-reviewed, ethically approved, written in English only and full text were included in the review. All descriptive, case-report, qualitative studies, literature reviews, study protocols and conference abstracts were excluded from the review process and duplicates from different searches were removed.

## 2.4 Type of interventions

The reviewed articles that used Motivational Interviewing (MI) as an intervention, delivered by trained professionals and targeted at Type 2 Diabetes Mellitus (T2DM) patients were clearly defined.

## 2.5 Types of outcome measures

Primary outcomes were glycosylated hemoglobin A1C (HgbA1C) value and depression. The measurement tools used to evaluate depression symptoms included were CES-D, Hospital Anxiety and Depression Scale Depression (HADS-D), Patient Health Questionnaire (PHQ-9), Beck Depression Inventory-II (BDI-II) and Generalized Anxiety Disorder Scale (GADS) [27–31].

## 2.6 Search methods (strategy) and review process

The databases of Cochrane Library, Google scholar, PubMed, MEDLINE Plus, ClinicalTrials. gov were searched systematically and gray literatures were reviewed for relevant studies based in the eligibility criteria. Key words and Mesh terms were used to search based on the PICOS model and included: "type 2 diabetes," "diabetes mellitus," "non-insulin dependent diabetes mellitus," "adult onset diabetes," "glycemic control," "glycosylated hemoglobin","HgbA1C", "Depression," T2DM, T2D, NIDD, NIDDM, "motivational interviewing," "MI," Motivation AND interviewing,"motivational interview". This search was completed in the 4th week of April 2019.

## 2.7 Selection of studies

All potentially relevant retrieved articles were investigated as full text. The inclusion and exclusion criteria for the papers included in the study were defined in the study protocol prior to commencing the review. Inclusion criteria (defined according to PICOS) were as follows: P: people with type 2 diabetes, I: Motivational Interviewing (MI) interventions, C: usual care or a non- Motivational Interviewing (MI) intervention, O: glycosylated hemoglobin A1C (HgbA1C) level and Depression and S: randomized controlled trials (RCTs) & pseudo RCT/ pre-post intervention. A single failed eligibility criterion is sufficient for a study to be excluded from the review. The study selection process was performed by two investigators (HB1, HB2, and K.K) independently. Differences in opinion were discussed. A detailed chart of the studies included and rejected was kept; this included reasons for rejection. The process of study selection is shown in (Fig 1).

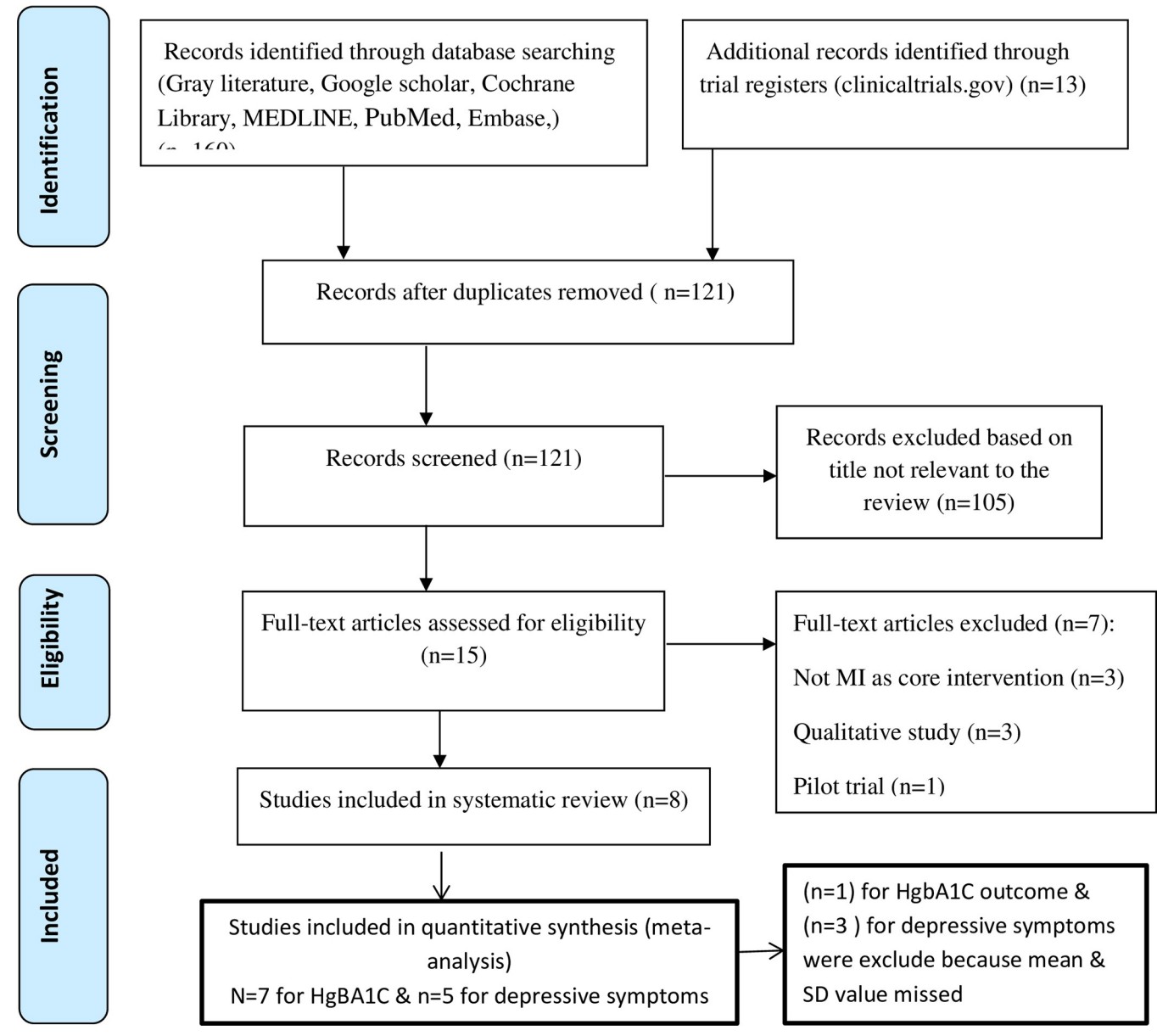

**Fig 1. PRISMA flow diagram of study retention process for the review.**

## 2.8 Assessment of methodological quality

After the process of study selection, the eight remaining articles were assessed by three investigators (HB1, HB2 and K.K) independently to determine their quality. Methodological quality of the studies was assessed using Joanna Briggs Institutes Critical Appraisal Checklists for randomized and pseudo-randomized studies (Tables 1 and 2). Articles with more than average score were included in the review.

## 2.9 Data extractions and syntheses

A detailed data extraction sheet was used to assist in the data extraction process for studies that fulfilled the inclusion criteria. The main variables extracted were: first author's name, year of

**Table 1. Evaluation of the methodological quality of included studies (RCT).**

| List of criteria | M. Chen et al. [39] | | Garry Welch et al. [16] | | Stuckey et al. [32] | | R.A. Gabbay et al. [33] | | K Ismail et al. [34] | | Kaltman S et al. [43] | |
|---|---|---|---|---|---|---|---|---|---|---|---|---|
| | yes | No | yes | No | yes | no | yes | No | yes | No | yes | No |
| Randomization | X | | X | | X | | X | | X | | X | |
| Allocation concealment | | X | | X | X | | | X | X | X | | X |
| Treatment group similar at base line | X | | X | | X | | X | | X | | X | |
| Participant blinding | X | | X | | X | | X | | X | | X | |
| Interventionist blinding | | X | | X | | X | | X | | X | | X |
| Outcome assessor blinding | | X | | X | | X | | X | | X | | X |
| treatment groups treated identically | X | | X | | X | | X | | X | | X | |
| follow up complete (Dropout & loss) | | X | | X | | X | | X | | X | | X |
| participants analysed in the groups | X | | X | | X | | X | | X | | X | |
| outcomes measured in the same way for treatment groups | X | | X | | X | | X | | X | | X | |
| outcomes measured in a reliable way | X | | X | | X | | X | | X | | X | |
| appropriate statistical analysis used | | X | | X | | X | X | | | X | X | |
| trial design appropriate | X | | X | | X | | X | | X | | X | |
| **Total score out of 13** | **8** | | **8** | | **9** | | **9** | | **9** | | **9** | |

Reviewed article Author's name 1, M. Chen et al. [39], 2. Garry Welch et al. [16], 3. Stuckey et al. [32], 4. R.A. Gabbay et al. [33], 5. K Ismail et al. [34] and 6. Kaltman S et al. [43].

publication, study design, number of participants at baseline and follow-up (sample size), eligibility criteria used, intervention delivery method (e.g. individual or group, face-to-face or remote delivery), intervention intensity (number of sessions, duration), health care staff who delivered the intervention and outcomes of the interventions (clinical and psychological measures), quality control (training, supervision, written manuals, and assessments of adherence or competence). Data were synthesized in a narrative way & meta-analysis was done.

**Table 2. Evaluation of the methodological quality of included studies (quasi experiment).**

| List of criteria | Celano et al. [44] | | Calhoun et al. [45] | |
|---|---|---|---|---|
| | yes | No | yes | No |
| Clear cause & effect | X | | X | |
| participants included in any comparisons similar | X | | X | |
| participants included in any comparisons receiving similar treatment/care | X | | X | |
| control group | | X | | X |
| multiple measurements of the outcome | X | | X | |
| Follow up completed | | X | | X |
| outcomes of participants included in any comparisons measured in the same way | X | | X | |
| outcomes measured in a reliable way | X | | X | |
| participants analysed in the groups | X | | X | |
| appropriate statistical analysis used | X | | X | |
| **Total score out of 10** | **8** | | **8** | |

Reviewed article Author's name: 1. Celano et al. [44] and 2. Calhoun et al. [45].

## 2.10 Dealing with missing data

Important numerical data such as screened, randomized participants as well as intention-to-treat, and as-treated and per-protocol populations were carefully evaluated. Attrition rates, e.g. drop-outs, losses to follow-up and withdrawals, and critically apprises issues of missing data and imputation methods (e.g. last observation carried forward) were investigated.

## 2.11 Subgroup analyses and investigations of heterogeneity

The following characteristics were expected to introduce heterogeneity, and subgroup analyses were done to investigate interactions: Short versus longer intervention duration (3 months versus 24 months), CES-D scale versus HADs or BDI-II scale, intervention session time/duration (30 minutes versus 60–80 minutes) and RCT versus Pre post intervention.

## 2.12 Sensitivity analyses

The strength of the results was tested by repeating the analysis using different measures of effect size (mean and SD.) and different statistical models (fixed-effect and random-effects models) and the pooled effect size of Motivational Interviewing (MI) was compared against all control groups. Sensitive article that have greater effect on the pooled effect estimate result was analyzed one by one

## 2.13 Statistical analyses

information extracted from the published articles was sample sizes at the baseline and follow-up in each group, mean and standard deviation at the baseline and follow-up in each group. Meta-analysis was conducted using RevMan 5.2.0 and STATA version 11 software. Data were evaluated by means of weighted mean differences (WMDs) and 95% confidence intervals (CI). Heterogeneity was explored by the I2 test. If I2 < 50%, a fixed-effects model was employed; otherwise, a random-effects model was used. Subgroup analysis was performed according to the heterogeneous factors (study design and length of follow-up period) considered. Potential publication bias was assessed using Egger's tests.

# 3. Results

## 3.1 General descriptions of studies

The detailed literature search process and rejection rationale are illustrated in the PRISMA flow diagram in Fig 1. The initial search revealed 173 articles from the databases. After removing duplicates, 121 studies were retained for further screening. The next steps involve analysis of titles then 105 records were excluded because the titles were not relevant to the review. Then a total of 15 full articles were screened and 7 studies were excluded due to they did not meet the inclusion criteria for this review. Finally, eight full articles were retained for the systematic review, seven articles and five articles were included in meta-analysis of glycosylated hemoglobin A1C (HgbA1C) and depressive symptoms outcome respectively (Fig 1).

All articles included in the review and analysis were published between the years 01/01/2009 and 12/06/2020 and were original, empirical studies. Six of the reviewed studies were RCTs [16,32–36] and two were quasi-experimental used pre-post a study design [37,38]. The main aims of all studies were to examine and assess the effect of Motivational interviewing (MI) on diabetes outcomes in people with type 2 diabetes mellitus and the quality of the studies was considered to be good (Tables 1 and 2). However, one out of the eight studies failed to provide appropriate information about methods used.

Although type of participants were similar, there were variations in the study designs used, how the intervention was delivered, who delivered the intervention, the number of sessions, length of each session, follow-up period and how intervention effects were examined and the setting (single vs. multi). Of the total eight eligible articles in systematic review one article was excluded from the review process because it failed to provide relevant information on effect of Motivation interviewing (MI) on depression and in meta-analysis two articles (Analysis on effect of Motivational Interviewing (MI) on glycosylated hemoglobin A1C (HgbA1C) and three articles (analysis on effect of Motivational Interviewing (MI) on depressive symptom) were excluded because Mean or SD or both were not found to allow it to be fully used in the analysis (Table 3).

The time taken for follow-up, small sample sizes, small number of intervention sessions and short time taken for each intervention session are worth discussing in relation to the articles included in this review. In four studies [39,43–45] the follow-up measurements for glycosylated hemoglobin A1C (HgbA1C) and depressive symptoms were done after only three months which can be considered as relatively short time to monitor changes in glycosylated hemoglobin A1C (HgbA1C) and depressive symptoms (Table 3).

All studies included descriptions of the inclusion and exclusion criteria, randomization technique (i.e. stratified permuted block randomization and random number generation) was

**Table 3. Summery of reviewed Studies evaluating effect of MI on Hgb.** A1C value and depression in people with T2DM.

| Name of first author/year of publication | Design, Settings | Sample size | Intervention/ session | Follow up (months)/ duration (mint) | Clinical indicators | Behavioral or psychological targets |
|---|---|---|---|---|---|---|
| S.M. Chen et al. (2012) [39] | RCT; single site | 215 T2DM patients | I: MI, C:DM health education/ 3 session | 3/40-60 mint. | HbA1c | DSM, Depression, Anxiety, Stress; DMSE and Quality of life-brief |
| Garry Welch et al.(2010) [20] | RCT; multi-site | 234 T2DM patients with HbA1c > 7.5%) | I: MI, C: DSME/3 sessions | 6/30 mint. | HbA1c, Body mass index | Diabetes distress, Diabetes self-care behaviors, Diabetes Treatment Satisfaction, Depression, DSM-Self-efficacy |
| Stuckey et al. (2009) [40] | RCT; multi-site | 549 T2DM patients | I: MI, C: Usual care/7 sessions | 24/60 mint. | HbA1c, BP, LDL, BMI | Emotional Distress, Treatment Satisfaction, Depression, self-care activities; and physician satisfaction |
| R.A. GABBAY et al.(2013) [41] | RCT; multi-site | 545 T2DM patients | I: MI C: Usual care/7 sessions | 24/60 mint. | HbA1c, BMI, LDL, SBP, DBP, glucose score | Emotional distress, Treatment satisfaction, Depression, Self-care, General diet score, Specific diet score, Exercise score, Foot score, quality of life |
| Ismail K et at. (2018) [42] | RCT; multi-site | 333 T2DM patients with persistent HbA1c ≥69.4 mmol/mol | I: MI, C: Usual care/12 sessions | 18/30 mint. | HbA1c, Systolic and diastolic blood pressure, BMI, waist circumference | Depressive symptoms, harmful alcohol intake, diabetes-specific distress, and cost-effectiveness. |
| Huang CY et al. (2016) [43] | RCT; single site | 61 T2DM patients | I: MI, C: usual care/12 session | 3/80 mint. | HbA1C, FBG, BMI | depressive symptoms, and both physical and mental quality of life |
| Celano et al. (2019) [44] | Pseudo-RCT (quasi experiment; single site | 20 T2DM patients with HbA1c > 6.5% or fasting glucose > 126 mg/dL),& | I: MI/9+ session | 3/30 mint. | HbA1C, BMI | Positive affect, Optimism, Anxiety, Depression, Dietary adherence, Diabetes-related adherence, Medication adherence, Self-reported activity |
| Calhoun et al. (2010) [45] | Pseudo-RCT (quasi experiment; single site | 26 T2DM patients | I: MI/3 sessions | 3/30 mint. | HbA1C, random blood glucose | Unhealthy dietary choices, Depression, DC Fatalism, Diabetes Locus of Control, diabetes quality of life and physical exercise |

*T2DM: Type 2 diabetes mellitus, RCT: Randomized control trial, MI: Motivational interviewing, DSME: Diabetes self-management education.

reported in three out of 6 RCTs and sample size was ranged from 20 [44,45] to 549 [40] (Table 3). The age range of the participants was 18–87 years of age. All studies defined person with type 2 diabetes and 18 and above years of age as inclusion criteria and person with severe co-morbidity, terminal illness and cognitive impairment as exclusion criteria.

### 3.2 Study characteristics and evaluation methods

Patient blood sample tests, were the objective measures used to measure clinical outcomes (HgbA1C, RBS or FB, lipid profile) [20,39–45], followed by validated and reliable question-naires to measure behavioral and psychological outcomes. (The Diabetes Self-Management, Diabetes Management Self-Efficacy, Quality of Life-brief, depression, anxiety and diabetes related distress and diabetes treatment satisfaction) [20,39–45] (Table 3).

### 3.3 Study participants and study settings

All studies focused exclusively on patients with the type 2 diabetes [20,39–45]. Four were con-ducted in primary care settings [40–42,44], three studies done in hospital based diabetes/endo-crinology outpatient clinics [39,20,43] and one study was done among American Indians in Indian health services (HIS) clinic [45]. Five studies were conducted in USA [20,40,41,45], two studies conducted in Taiwan [39,43] and only one study done in UK [42].

### 3.4 Description of interventions

The way Motivational interviewing (MI) used as a behavioral intervention varied between the studies included in this review, Table 2 provides a summary. Notable differences were shown among the studies especially in the key components associated with tailoring the MI interven-tion such as the professional who delivered it, the number of sessions, time spent per session and follow-up period. Professionals from different fields delivered the intervention in the reviewed studies.

Three of the studies used nurse case manager as interventionists who received Motivational interviewing (MI) training [39,40–42]. In one study all the participants [20] received Motiva-tional Interviewing (MI) from two certified diabetes educators; the diabetic educators had received the training about Motivational Interviewing (MI) from two experienced trainers. Two studies [43] used psychotherapist, one study [44] used psychologist and only one study [45] used both diabetes educators and behavioral health specialists to deliver the Motivational Interviewing (MI) intervention (Table 3).

There studies had notable variations in how Motivational interviewing (MI) sessions were constructed and how many sessions were offered to their participants. The number of sessions offered ranged from three [20,39,45] to twelve [42,43]. In two of the studies [40,41], Motiva-tional Interviewing (MI) was given to the participants for seven sessions throughout the stud-ies. In four studies [20,42,44,45] participants received Motivational Interviewing (MI) for about 30 minutes per session. In three studies [39–41] participants received MI for 60 minutes and in another study [43] participants received Motivational Interviewing (MI) sessions of 80 minute durations (Table 3).

### 3.5 Assessment of risk of bias in included studies

Risk of bias was assessed using the Cochran Collaboration's tool and the criteria for risk of bias was judged as 'low risk', 'high risk' or 'unclear risk' and individual bias items was used as described in the Cochran Handbook for Systematic Reviews of Interventions [39] (Table 4).

**Table 4. Risk of bias assessment for included studies (Cochrane method).**

| Source | Random sequence generation (selection bias) | Allocation concealment (selection bias) | Blinding of participants & personnel (performance bias) | Blinding of outcome assessment (detection bias) | Incomplete outcome data (attrition bias) | Reporting bias | Other bias |
|---|---|---|---|---|---|---|---|
| S.M. Chen et al. [40] | ? | ? | + | + | + | + | + |
| Garry Welch et al. [21] | ? | ? | + | + | + | + | + |
| Stuckey et al. [41] | + | + | ? | + | + | – | + |
| R.A. GABBAY et al. [42] | ? | ? | ? | + | + | + | + |
| Ismail K et at. [43] | + | + | + | + | + | + | + |
| Huang CY et al. [44] | + | ? | + | + | + | + | + |
| Celano et al. [45] | – | – | – | + | + | + | + |
| Calhoun et al. [46] | – | – | – | + | + | + | + |

'+': Low risk of bias in study design, '-': High risk of bias in study design, '?': Unclear or insufficient detail.

## 3.6 Effect of interventions

The effect of the Motivational interviewing (MI) intervention was measured in different ways in the eight reviewed studies. All studies [20,39–45] included outcome measures or evaluation of change in patients' glycosylated hemoglobin A1C (HgbA1C), BMI and depressive symptoms. Five studies [20,39–42] measured diabetes related distress, six studies [20,39–41,44,45] measured diabetes self-care activities (diet, exercise, home glucose monitoring and medication adherence). Five studies [20,39,41,43,45] evaluate diabetes related quality of life. The reviewed studies also measured self-regulation [39], Diabetes Management Self Efficacy [20,39], Cost effectiveness [40,42], Diabetes Treatment Satisfaction [20,40,41], alcoholism [42], DC Fatalism, Diabetes Locus of Control [45] (Table 5).

**Table 5. Selected behavioral, psychological and clinical outcomes based on tests of significance between the Intervention and control group.**

| Source | Clinical target | | | | Behavioral target | | | | Psychological targets | | |
|---|---|---|---|---|---|---|---|---|---|---|---|
| | A1C | BMI | BP | Cholesterol level | Diet | Physical activity | SDMA | DSM efficacy | Depressive symptoms | Distress | DrQL |
| S.M. Chen et al.(2012) [35] | + | 0 | 0 | 0 | 0 | 0 | + | + | + | 0 | + |
| Garry Welch et al.(2010) | - | 0 | 0 | 0 | 0 | 0 | 0 | 0 | 0 | 0 | 0 |
| Stuckey et al. (2009) [32] | Ns | Ns | Ns | Ns | 0 | 0 | 0 | 0 | Ns | Ns | Ns |
| R.A. GABBAY et al.(2013) [33] | Ns | 0 | + | Ns | 0 | 0 | Ns | 0 | + | Ns | Ns |
| Ismail K et at.(2018) [34] | Ns | Ns | Ns | Ns | 0 | 0 | 0 | 0 | Ns | 0 | 0 |
| Huang CY et al.(2016) [36] | + | + | 0 | 0 | 0 | 0 | 0 | 0 | + | 0 | + |
| Celano et al.(2019)[37] | Ns | Ns | 0 | 0 | Ns | + | + | 0 | Ns | 0 | 0 |
| Calhoun et al.(2010) [38] | Ns | 0 | 0 | 0 | Ns | + | 0 | 0 | + | 0 | + |

Note. 0 = not measured/reported, NS = non-significant outcomes, + = statistically significant positive outcomes,— = significant difference in favor of the control group, BMI: Body Mass Index, SMBG: Self-Monitoring of blood glucose, DSM: Diabetes self-management, DrQL: Diabetes related quality of life.

## 3.7 Effect of Motivational Interviewing (MI) on glycosylated hemoglobin A1C (HgbA1C)

Three out of eight reviewed studies [39,43,44] found that Motivational interviewing (MI) resulted in a significant improvement of mean glycosylated hemoglobin A1C (HgbA1C) in the Intervention group when compared with the control group. Two studies [20,41] showed a mean glycosylated hemoglobin A1C (HgbA1C) level improvement in both group. In one study [20] the mean change of glycosylated hemoglobin A1C (HgbA1C) of Control group was greater than intervention group. Two studies [42,45] indicated that there was no effect of intervention on mean change of glycosylated hemoglobin A1C (HgbA1C) in either group. One study [40] do not clearly describe about the mean change of glycosylated hemoglobin A1C (HgbA1C) due to the effect of MI was not clearly described (Table 5).

## 3.8 Effect of Motivational Interviewing (MI) on depressive symptoms

Four reviewed studies identified that there was statistically significant improvement of depressive symptoms among the participants. Which is in one study [39] the change was observed in pre and post intervention. A statistically significant decrease in depressive symptoms was found in three studies [41,43,45] in post intervention when compared with symptoms at baseline. Three studies [40,42,44] described small change or improvement in depressive symptoms were seen in post intervention when compared with symptoms at baseline but not statistically significant. In another study [20] the direct effect of the intervention on depression was not assessed but mediator effect of depression on glycosylated hemoglobin A1C (HgbA1C) was investigated (Table 5).

## 3.9 Results of meta-analysis

**3.9.1. Effects of Motivational Interviewing (MI) on glycosylated hemoglobin A1C (HgbA1C) level.** A total of eight studies [20,39,41–45] was reviewed and reported evaluation of the effects of MI on the HbA1c value. One trial [42] was excluded from the analysis because Mean and SD of HgbA1C was not reported. A subgroup analysis was not performed because of the heterogeneity test result showed that an $I^2$ of 38% which is <50%; thus, the random-effects model was not used. Therefore, the pooled result showed that the HgbA1c level was significantly lower in the intervention group than in the control group (WMD, -0.27; 95% CI, -0.46 to -0.09; p = 0.004) (Fig 2).

**3.9.2 Sensitivity analysis and publication bias.** Each article was checked for sensitivity and one article [43] was the most sensitive because the $I^2$ become 0% from 38% and pooled effect estimate result become -0.20 [-0.39, 0.03] from -0.27 [-0.46, -0.09] (Table 6). The publication bias was determined using Egger's statistical tests. The test result showed that there was no publication bias (P = 0.393).

**3.9.3. Effects of Motivational Interviewing (MI) on depressive symptoms.** Of eight reviewed studies five articles [40,41,43–45] reported assessment of the effects of Motivational Interviewing (MI) on the depressive symptoms. Three studies [20,39,42] were excluded because Mean, SD or both were not reported. The heterogeneity test result revealed an $I^2$ of 93% which is >50% (Fig 3); thus, a random-effects model was used to perform subgroup analysis according to MI session time (30 or 60–80 minutes) and Follow-up period (3 or 24 months).

**3.9.3.1. Effect of 30 minute Motivational Interviewing (MI) session time:** Two studies [44,45] assessed the effect of Motivational Intervening (MI) on depressive symptoms using pre-post as study design or intervention session time of 30 minutes. The Random effect model

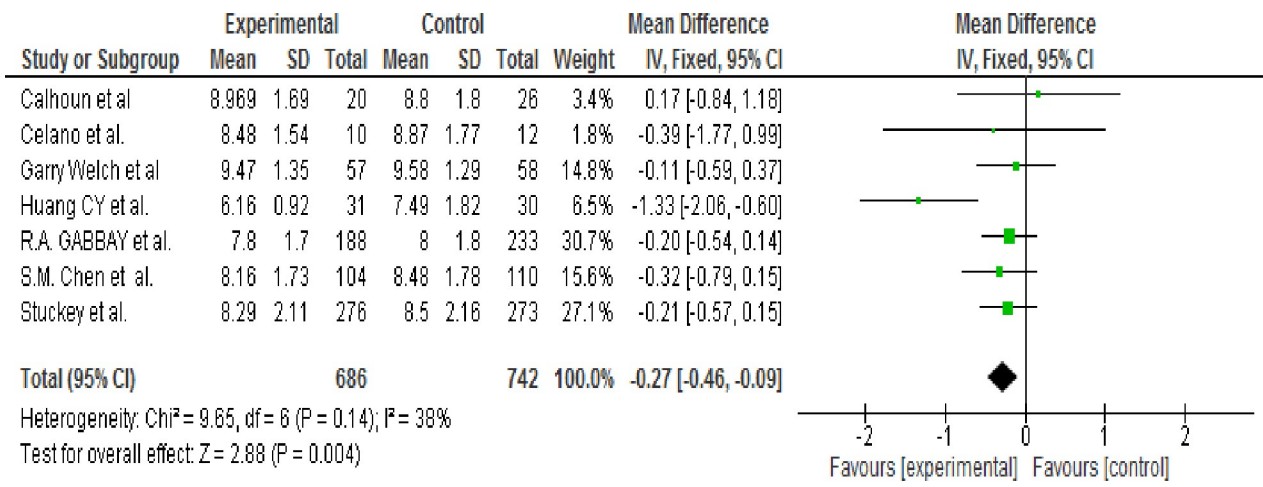

**Fig 2. Efficacy of motivation interviewing for reduction of HbA1c (fixed-effects model).**

analysis result showed that depressive symptoms were not significantly decreased in the intervention group when compared with the control group (WMD, -1.58; 95% CI, -5.05 to -0.188; p = 0.37; $I^2$ = 48%) (Fig 4).

**3.9.3.2. Effect of 60 minute Motivational Interviewing (MI) session time:** Three studies [40,41,43] assessed the effect of Motivational Interviewing (MI) on depressive symptoms using RCT as a study design or intervention session time of 60–80 minutes. The Random effect model analysis result showed that there was no significant difference in the reduction of depressive symptoms between the intervention and control groups (WMD, -4.30; 95% CI, -9.32 to -0.73; p = 0.09; $I^2$ = 95%) (Fig 5).

**3.9.3.3 Effect of 3 months of Motivational Interviewing (MI) follow up period:** Three studies [43,44,45] assessed the effect of three months duration of Motivational Interviewing (MI) on depressive symptoms. At the end of the study period depressive symptoms had not significantly decreased in the intervention group when compared the control group *(WMD, -4.45; 95% CI, -10.58 to 1.69; p = 0.16; $I^2$ = 96%)* (Fig 6).

**3.9.3.4 Effect of 24 months of MI follow up period:** Two studies [40,41] assessed the effect of 24 months duration of Motivational (MI) on depressive symptoms. No significant difference in the reduction of depressive symptoms was observed between the intervention and control groups *(WMD, -2.12; 95% CI, -5.54 to 1.30; p = 0.22; $I^2$ = 83%)* (Fig 7).

**3.9.3.5 Sensitivity analysis and publication bias:** Each article was checked for sensitivity. One article [43] was the most sensitive article because the $I^2$ become 63% from 38% and pooled

**Table 6. Sensitivity test comparison of articles (effect of MI on Hgb. A1C).**

| Authors' name & year of publication | Total pooled results | 95% CI | I2 in % | P-value |
|---|---|---|---|---|
| Calhoun et al. 2010 [38] | -0.29 | [-0.48, -0.10] | 44 | 0.11 |
| Celano et al. 2019 [37] | -0.27 | [-0.46, -0.08] | 48 | 0.09 |
| Garry Welch et al. 2010 | -0.30 | [-0.50, -0.10] | 48 | 0.10 |
| Huang CY et al. 2016 [36] | -0.20 | [-0.39, -0.01] | 0 | 0.96 |
| R.A. GABBAY et al. 2013 [33] | -0.31 | [-0.53, -0.08] | 47 | 0.09 |
| S.M. Chen et al. 2012 [35] | -0.26 | [-0.47, -0.06] | 48 | 0.09 |
| Stuckey et al. 2009 [32] | -0.30 | [-0.51, -0.08] | 47 | 0.09 |

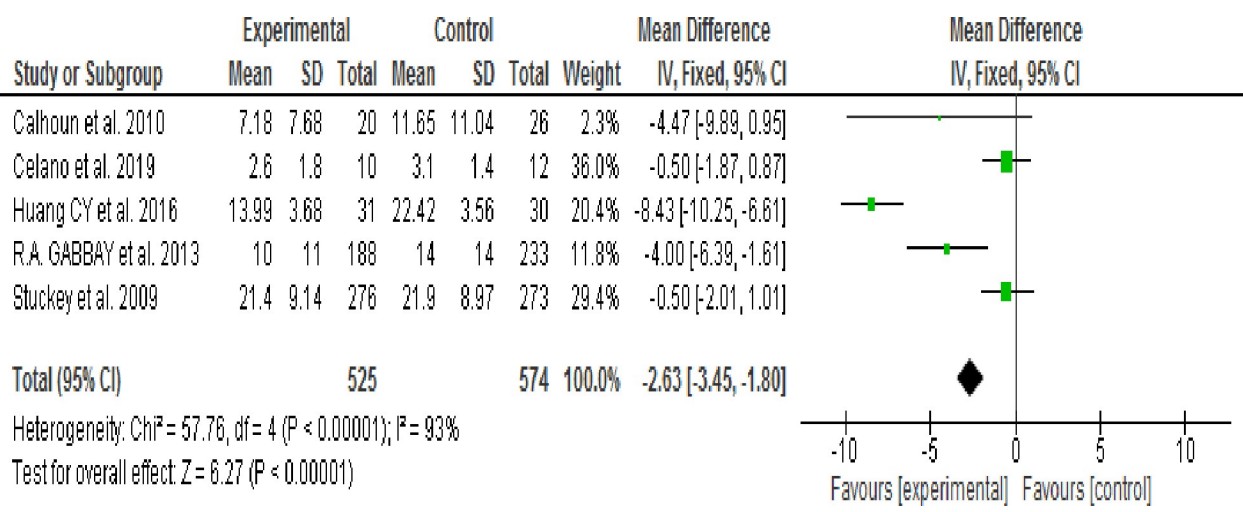

**Fig 3. Effect of MI to decrease depressive symptoms (fixed-effects model).**

effect estimate result become -1.14 [-2.06, 0.21] from -2.63 [-3.45, -1.80]. The publication bias was determined using Egger's statistical test. Egger's test result showed that there was no publication bias (P = 0.238) (Table 7).

## 4. Discussion

It is important to discover the effect of different types of psychological interventions on patients' clinical, behavioral and psychological outcomes because of the increase in prevalence of T2D, diabetes complication and health care costs. The aim of this review was to identify the effect of MI on HgbA1c value and depression in people with T2D.

Motivational interviewing (MI) is a relatively new and inspiring method for the development and improvement of the patients' therapeutic commitment [40]. Findings show that features of motivational interviewing (MI) help clients to engage in their treatment and build their consistency [41]. It is the best technique for helping clients who struggle with behavioral changes [42]. A review study in China reported Motivational Interviewing (MI) was the most effective psychological intervention for type 2 diabetes [43]. Two systematic reviews and one meta-analysis revealed that Motivational Interviewing (MI) interventions showed promising results for dietary behaviors and weight management in people with T2D [5], likely reduces elevated diabetes related distress [44], improved self-management abilities among patients

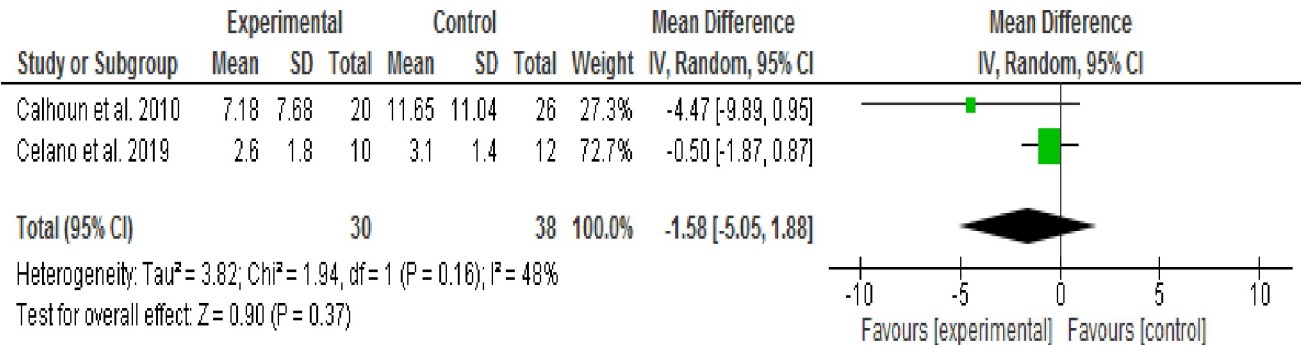

**Fig 4. Effect of MI on depressive symptoms using 30 mins session time (random-effects model analysis result).**

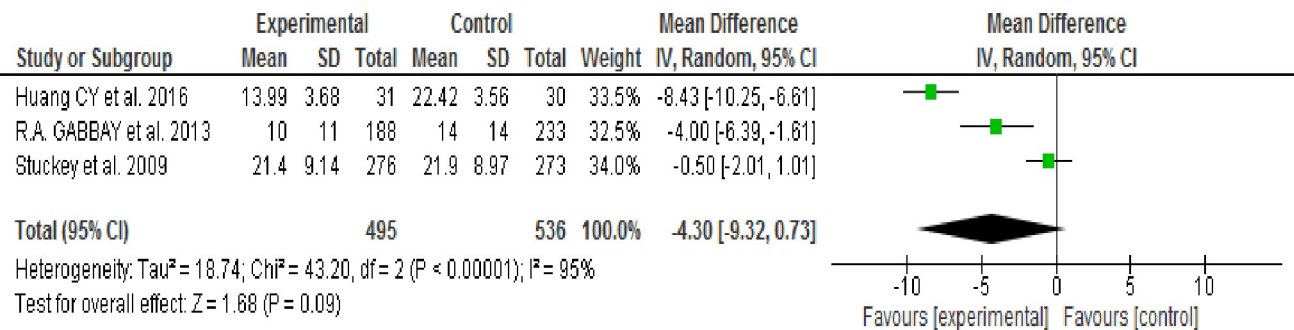

**Fig 5. Effect of MI on depressive symptoms 60–80 minutes session time (random-effects model analysis result).**

with type 2 diabetes, and short-term Motivational Interviewing (MI) intervention (<6 months) decrease glycosylated hemoglobin A1C (HgbA1C) effectively [27].

The pooled meta-analysis result showed that glycosylated hemoglobin A1C (HgbA1C) level was significantly lower in the intervention group when compared with the control group. However, research finding showed that glycosylated hemoglobin A1C (HgbA1C) reduction was observed in both groups [45] and a number of qualified experimental studies evidenced that client who takes Motivational Interviewing (MI) shows improvement on self-management as well as manifestations of illness [46].

Pooled estimate result using fixed effect model showed that depressive symptoms were significantly decreased in the intervention group when compared with the control group and heterogeneity was significant. Therefore, subgroup analysis was performed using a random-effects model according to study type (RCT & Pseudo RCT), MI intervention duration (Shorter/3 months Vs. Longer/24 months) and MI intervention session time (30 minutes Vs. 60–80 minutes).

The result indicated that depressive symptoms were not significantly lower in the intervention group when compared with control group. However, evidence indicated that psychological therapies like Motivational Interviewing (MI) may also be effective in treating depression in people with diabetes but may have limited effects on glycemic outcomes [47,48]. Two possible reasons for these findings may be considered. First, there is greater heterogeneity among reviewed studies. Second as found by a previous study [49], the positive effects of diabetes education may gradually weaken with time (3 months Vs. 24 months).

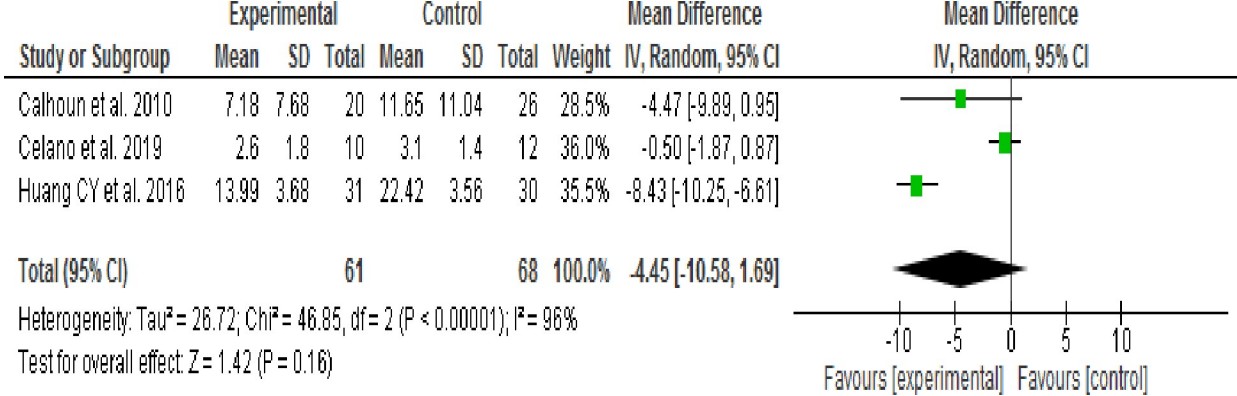

**Fig 6. Effect of MI on depressive symptoms at 3 months follow-up period (random-effects model analysis result).**

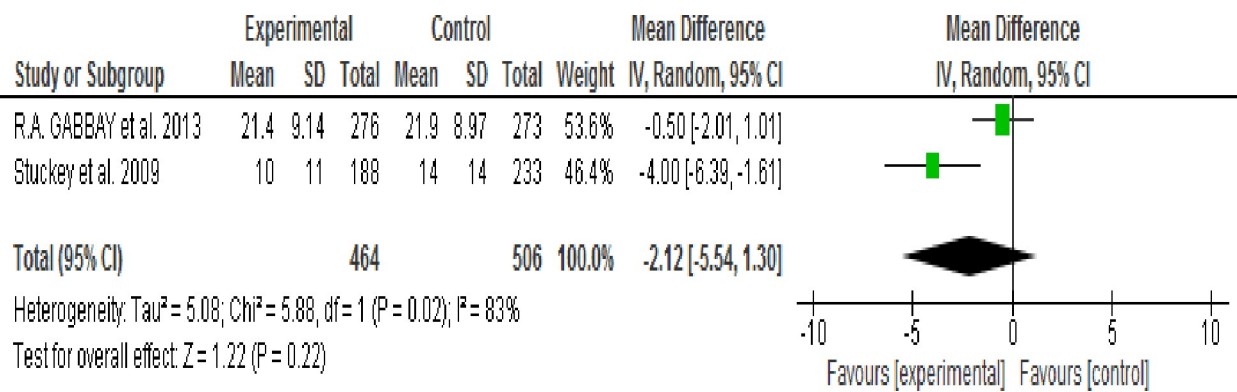

**Fig 7. Effect of MI on depressive symptoms at 24 months follow-up period (random-effects model analysis result).**

## 5. Limitations

The following limitations occurred while conducting this systematic review and meta-analysis:

- Because of strict implementation of selection criteria of published articles, the number of articles or studies included in the review limits to eight out of one hundred twenty-one searched articles.

- Very limited published articles specially study done in developing country countries.

- The studies were considerably different from one another regarding contents and quality this may result in increased heterogeneity.

- Only studies published in English were included in this review & analysis which introduced a risk of language bias: it is possible that studies reporting statistically significant results in the field have been published in other languages.

- There might be possibility of publication bias because of Authors' tendency to publish only reports of trials that have produced positive results.

## 6. Conclusion

The pooled result in meta-analysis indicated that Motivational Interviewing (MI) intervention is effective in reducing glycosylated hemoglobin A1C (HgbA1C) but not depressive symptoms of patients with type 2 diabetes. Further studies are needed to examine in particular high-quality quasi experiment or RCTs with large samples, studies assessing effect of Motivational Interviewing (MI) on psychological outcomes (diabetes related distress, depression and anxiety/stress.), appropriate number of Motivational Interviewing (MI) sessions and follow-up period which are required to increase the evidence in support of the advantages of Motivational

**Table 7. Sensitivity test comparison of articles (effect of MI on depression symptoms).**

| Authors' name & year of publication | Total pooled results | 95% CI | I2 in % | P-value |
|---|---|---|---|---|
| Calhoun et al. 2010 [38] | -2.58 | [-3.41, -1.75] | 95 | <0.00001 |
| Celano et al. 2019 [37] | -3.82 | [-4.85, -2.80] | 93 | <0.00001 |
| Huang CY et al. 2016 [36] | -1.14 | [-2.06, -0.21] | 65 | 0.04 |
| R.A. GABBAY et al. 2013 [33] | -2.44 | [-3.32, -1.57] | 95 | <0.00001 |
| Stuckey et al. 2009 [32] | -3.51 | [-4.49, -2.53] | 94 | <0.00001 |

Interviewing (MI). Finally, although qualitative materials are very important in public health research, very few such studies were found on this subject. Qualitative studies are needed to ascertain the benefits of Motivational Interviewing (MI) over traditional intervention and provide higher-quality study content with more accurate results

## Supporting information

**S1 File. PRISMA 2009 Checklist.**
(DOC)

**S2 File.**
(ZIP)

## Acknowledgments

We would like to acknowledge Dr. Mary Moran and Dr. Carmine for their contribution in English language (usage, spelling and grammar) edit of our manuscript.

## Author Contributions

**Conceptualization:** Kalayou Kidanu Berhe.

**Formal analysis:** Kalayou Kidanu Berhe.

**Methodology:** Kalayou Kidanu Berhe.

**Supervision:** Haftu Berhe Gebru, Hailemariam Berhe Kahsay.

**Writing – original draft:** Kalayou Kidanu Berhe.

**Writing – review & editing:** Kalayou Kidanu Berhe, Haftu Berhe Gebru, Hailemariam Berhe Kahsay.

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
