## [Decision Letter · Decision Letter 0]

6 Aug 2020

PONE-D-20-12859

EFFECT OF MOTIVATIONAL INTERVIEWING INTERVENTION ON HGB A1C VALUE AND DEPRESSION IN PEOPLE WITH TYPE 2 DIABETES MELLITUS (T2DM) (systematic review & meta-analysis)

PLOS ONE

Dear Dr. Berhe,

Thank you for submitting your manuscript to PLOS ONE. After careful consideration, we feel that it has merit but does not fully meet PLOS ONE’s publication criteria as it currently stands. Therefore, we invite you to submit a revised version of the manuscript that addresses the points raised during the review process.

We look forward to receiving your revised manuscript.

Kind regards,

Cheng Hu

Academic Editor

PLOS ONE

Journal Requirements:

2.

We suggest you thoroughly copyedit your manuscript for language usage, spelling, and grammar. If you do not know anyone who can help you do this, you may wish to consider employing a professional scientific editing service.  

4. Thank you for including your funding statement; none

5. Thank you for including your competing interests statement; "none"

6. Please upload a copy of Figure 7, to which you refer in your text on page 27. If the figure is no longer to be included as part of the submission please remove all reference to it within the text.

Reviewers' comments:

Reviewer's Responses to Questions

**Comments to the Author**

1. Is the manuscript technically sound, and do the data support the conclusions?

Reviewer #1: Yes

2. Has the statistical analysis been performed appropriately and rigorously? 

Reviewer #1: Yes

3. Have the authors made all data underlying the findings in their manuscript fully available?

Reviewer #1: Yes

4. Is the manuscript presented in an intelligible fashion and written in standard English?

Reviewer #1: Yes

5. Review Comments to the Author

Reviewer #1: Kalayou K Berhe et al. performed a systemic review and meta-analysis to evaluate the effectiveness of motivational interviewing intervention on glycemic control and on depressive symptoms for patients with diabetes. This study encompasses an interesting topic, but it has some critical problems.

1. The aim of study is to reveal the effect of motivational interviewing intervention on depressive symptoms for patients with diabetes. However, author’s assessment for them are unclear. In the result section, they claimed depressive symptoms was improved by motivational interviewing intervention. Cobversely, they concluded that motivational interviewing intervention is not effective for depressive symptoms.

2. The manuscript needs to be edited.

Minor comments

1. Hgb. A1C is unusual abbreviation. Motivation interviewing should be expressed as full spelling.

2. What is the difference for NIDD and NIDDM?

3. I feel difficulty to see Table 1. Please reconsider to the structure of table1.

4. Fig2, 3; Font type should be changed as the same as the others.

5. Table 6; The font type for “Huang” is different for the others.

6. Ref 34 seems to include Chinese characteristics.

7.

Comments to the Editor

Authors answered all comments, which I provided, and revised the manuscript well.

6. PLOS authors have the option to publish the peer review history of their article (what does this mean?). If published, this will include your full peer review and any attached files.

Reviewer #1: **Yes: **Masahide Hamaguchi

---

## [Author Response · Author response to Decision Letter 0]

14 Sep 2020

1. The aim of study is to reveal the effect of motivational interviewing (MI) intervention on depressive symptoms for patients with diabetes. However, author’s assessment for them is unclear (miss match between result and conclusion).

• The finding of four studies in systematic review and pooled result of meta-analysis (fixed effect model analysis) identified that there was statistically significant improvement of depressive symptoms

• But the pooled result of meta-analysis in fixed effect model analysis had heterogeneity then further subgroup analysis using random effect model was required then to get the true effect subgroup analysis according to MI session time and follow-up period was done. 

•There for pooled result of subgroup analysis showed that MI intervention is not effective for depressive symptoms (means do not have significant difference in the reduction of depressive symptoms among intervention and control group) 

• Then to avoid confusion results of systematic review and pooled result in fixed effect model analysis for depressive symptoms were removed. 

• The pooled results of subgroup analysis rephrased in line to conclusion and in such away readers could understand as per your comment.

• Then Rephrased as “ Effect of MI intervention on depressive symptoms was identified through subgroup analysis according to intervention session time (30 or 60-80 minutes) and Follow-up period (3 or 24 months) then result showed that there was no significant difference in the reduction of depressive symptoms between the intervention and control groups. The output results were (WMD, -1.58; 95% CI, -5.05 to -0.188; p = 0.37; I2=48%), (WMD, -4.30; 95% CI, -9.32 to -0.73; p = 0.09; I2=95%) , (WMD, -4.45; 95% CI, -10.58 to 1.69; p= 0.16; I2=96%) and (WMD, -2.12; 95% CI, -5.54 to 1.30; p = 0.22; I2=83%) respectively”. page 2 and 3 (Abstract ) and page 25

2. HgbA1C is unusual abbreviation. Motivation interviewing should be expressed as full spelling.

• HgbA1C is written in usual way or expanded form as glycosylated hemoglobin A1C (HgbA1C) throughout the manuscript as per your comment and MI is also written in the expanded form as Motivation interviewing (MI) throughout the manuscript as per your comment 

3. What is the difference for NIDD and NIDDM ( Search strategy section: on page 7 and 8 )

• Both are the same used to describe type 2 diabetes mellitus as none insulin dependent diabetes or diabetes mellitus and there is no difference. Those are key words or mesh terms we used to search articles for review because authors may use either of them interchangeably in their article as key word.

4. I feel difficulty to see Table 1. Please reconsider to the structure of table1. 

• Table 3 (page 15& 16) font size increased or corrected to make it visible and Table 1 (Page 9) “Evaluation of the methodological quality of the reviewed studies (RCT) (6 in number) based on JBI appraisal checklist. Its column is reviewed articles name of first author (with reference number: 39,16,32,33, 34 and 43) and its raw is evaluation criteria (yes/No). Font size increased to make it visible 

5. Fig2, 3; Font type should be changed as the same as the others.

• Because of the figures are analyses outputs of the software in the form of picture then copied and directly pest directly because of this could not change or modify the font style except increase its size. Therefore as the size of the picture or figure increase then font size also increased so that legible to the reader. Additionally the main important pooled result were written properly as (WMD, -0.27; 95% CI, -0.46 to -0.09; p= 0.004) and (WMD, -2.63; 95% CI, -3.45 to -1.80; p< 0.00001) respectively (Page 19 & 20). Moreover, as requirement of the journal figures are uploaded separately & individually in the form of TIFF

6.Table 6; the font type for “Huang” is different for the others. 

• The font type for “Huang” which is differ for the others or made bold then corrected or made similar font type like others or avoid from making bold the word Huang (page 20 & 22). 

7.Ref 34 seems to include Chinese characteristics: The Chinese character is removed (Reference section)

---

## [Decision Letter · Decision Letter 1]

5 Oct 2020

Effect of Motivational Interviewing Intervention on HgbA1C and Depression in people with Type 2 Diabetes Mellitus: A systematic review and meta-analysis

PONE-D-20-12859R1

Dear Dr. Berhe,

We’re pleased to inform you that your manuscript has been judged scientifically suitable for publication and will be formally accepted for publication once it meets all outstanding technical requirements.

Kind regards,

Cheng Hu

Academic Editor

PLOS ONE

Additional Editor Comments (optional):

Reviewers' comments:

Reviewer's Responses to Questions

**Comments to the Author**

1. If the authors have adequately addressed your comments raised in a previous round of review and you feel that this manuscript is now acceptable for publication, you may indicate that here to bypass the “Comments to the Author” section, enter your conflict of interest statement in the “Confidential to Editor” section, and submit your "Accept" recommendation.

Reviewer #1: All comments have been addressed

2. Is the manuscript technically sound, and do the data support the conclusions?

Reviewer #1: Yes

3. Has the statistical analysis been performed appropriately and rigorously? 

Reviewer #1: Yes

4. Have the authors made all data underlying the findings in their manuscript fully available?

Reviewer #1: Yes

5. Is the manuscript presented in an intelligible fashion and written in standard English?

Reviewer #1: Yes

6. Review Comments to the Author

Reviewer #1: I think that this manuscript has a benefit to publish and authors answered all comments, which I provided, and revised the manuscript well.

7. PLOS authors have the option to publish the peer review history of their article (what does this mean?). If published, this will include your full peer review and any attached files.

Reviewer #1: No

---

## [Editor Report · Acceptance letter]

7 Oct 2020

PONE-D-20-12859R1 

Effect of Motivational Interviewing Intervention on HgbA1C and Depression in people with Type 2 Diabetes Mellitus(Systematic review and Meta-analysis) 

Dear Dr. Berhe:

I'm pleased to inform you that your manuscript has been deemed suitable for publication in PLOS ONE. Congratulations! Your manuscript is now with our production department. 

Kind regards, 

on behalf of

Dr. Cheng Hu 

Academic Editor

PLOS ONE